# Effect of Wing Membrane Material on the Aerodynamic Performance of Flexible Flapping Wing

Xinyu Lang [1], Bifeng Song [1,2,3], Wenqing Yang [1,2,3] and Xiaojun Yang [1,2,3,*]

1 School of Aeronautics, Northwestern Polytechnical University, Xi'an 710072, China;
langxinyu@mail.nwpu.edu.cn (X.L.); sbf@nwpu.edu.cn (B.S.); yangwenqing@nwpu.edu.cn (W.Y.)
2 Research & Development Institute, Northwestern Polytechnical University in Shenzhen,
Shenzhen 518057, China
3 Yangtze River Delta Research Institute, Northwestern Polytechnical University, Taicang 215400, China
* Correspondence: xjyang518@nwpu.edu.cn

**Abstract:** Flexible deformation of the insect wing has been proven to be beneficial to lift generation and power consumption. There is great potential for shared research between natural insects and bio-inspired Flapping wing Micro Aerial Vehicles (FWMAVs) for performance enhancement. However, the aerodynamic characteristics and deformation process of the flexible flapping wing, especially influenced by wing membrane material, are still lacking in-depth understanding. In this study, the flexible flapping wings with different membrane materials have been experimentally investigated. Power input and lift force were measured to evaluate the influence of membrane material. The rotation angles at different wing sections were extracted to analyze the deformation process. It was found that wings with higher elastic modulus membrane could generate more lift but at the cost of more power. A lower elastic modulus means the wing is more flexible and shows an advantage in power loading. Twisting deformation is more obvious for the wing with higher flexibility. Additionally, flexibility is also beneficial to attenuate the rotation angle fluctuation, which in turn enhances the aerodynamic efficiency. The research in this paper is helpful to further understand the aerodynamic characteristics of flexible flapping wing and to design bio-inspired FWMAVs with higher performance.

**Keywords:** flapping wing; membrane material; flexible wing; wing deformation; bio-inspired

## 1. Introduction

Flying insects have drawn the attention of both biologists and engineers for their unsurpassed aerodynamic performance and maneuverability. Unsteady aerodynamic mechanisms of insect wings have been widely and deeply investigated [1,2], and are responsible for the high lift generation. Recently, with the application of high-speed photography technology, the flapping process of the insect wings has been clearly observed during free-flying movement [3]. The wing deformation due to flexibility, such as bending and twisting, was found to be an obvious phenomenon. Furthermore, wing deformation is prevalent among flying insects like hawkmoths, dragonflies, damselflies, etc. Recently, research on the influence of wing flexibility has shown that the flexible wing presents advantages in energy saving and lift generation [4,5].

In most insects, the wings' deformations are usually composed of bending, twisting, and camber changes [3,6]. The distribution of the wing veins makes the flexural stiffness in spanwise and chordwise directions very different [7]. The leading edge with thicker vein and membrane forms higher flexural stiffness along the spanwise direction, indicating that twisting deformation and camber change will be more obvious. According to Shumway et al. [8], who analyzed the flow around the deformed dragonfly wing, the camber change reduced flow separation on the outboard wing, thus improving the aerodynamic efficiency. Addo-Akoto et al. [9] experimentally compared the rigid and flexible wing, and the result shows the flexible wing could increase the lift generated at the end of the stroke by releasing

its stored elastic energy in the form of twisting and camber. Numerous studies have been performed on the investigation of the flexible flapping wing and the results have proved that flexibility has advantages in force generation and aerodynamic efficiency [10,11]. Nevertheless, Godoy-Diana and Thiria [12] suggested that there exists an optimum flapping frequency beyond which the efficiency starts to decrease.

In the past few years, the Flapping wing Micro Aerial Vehicles (FWMAVs), which mimic birds and insects, have attracted a lot of attention for their outstanding maneuverability and high efficiency. To design the FWMAVs with higher flight ability and aerodynamic performance, similar to natural flyers, is a persistent pursuit for the researcher. Therefore, numerous efforts have been conducted to design and optimize the flapping wing with higher aerodynamic characteristics, which play a critical role in the vehicles' performance [13–15]. Truong et al. [16] reported the development and characteristics of artificial wings that can reproduce camber and twisting deformations. Nan et al. [17] performed experimental optimization on the shape of flexible flapping wing with the test parameters including camber angle, aspect ratio, etc. Phan et al. [18] developed several deformable wings with different vein structures, and the results presented potential applications on the FWMAVs. Au et al. [19] and Nguyen et al. [20] innovatively proposed the design of a wing with distributed corrugations along the wingspan, while this arrangement would slightly augment the lift performance. Recently, Lee et al. [21] presented an experimental design optimization of a flapping wing via a surrogate-based global optimization method. Significant improvement in the lift efficiency confirmed the successful optimization process. Although there has been considerable progress in the investigation of flexible flapping wing for the FWMAVs, there is still a lack of research on the influence of some flexible parameters, such as the influence of wing membrane materials.

In this study, an experiment was conducted to investigate the aerodynamic characteristics of the deformable flapping wing with different flexibility. The flexibility of the flapping wing was adjusted by replacing the wing membrane with three different elastic moduli. A flapping mechanism was developed to generate flapping motion. The wing was driven to perform a flapping motion with passive rotation, accompanied by spanwise twisting deformation. Lift force and power consumption were measured to evaluate the performance of the different flexible wings. Wing deformation was captured using high-speed cameras. To analyze the influence of flapping frequency and membrane material, the rotation angle and angle of attack at various wing sections were extracted and compared. The present research will provide a better understanding of the aerodynamic and deformable characteristics of the flexible flapping wing and offer guidance for the selection of wing membrane material.

## 2. Materials and Methods

### 2.1. Flapping Mechanism

The flapping mechanism was designed based on the combination of crank-slide and rack-pinion mechanism. Figure 1a shows the schematic of the flapping wing mechanism. Figure 1b presents the assembled computer-aided design (CAD) model of this flapping mechanism. The reduction gear, driven by a DC motor, converts the rotational motion into the linear motion of the slider through the linkage. Racks installed on both sides of the slider convert linear oscillation into pinion gear rotational oscillation.

From Figure 1a, the relationship of the linkages can be easily derived and expressed as follows:

$$h = l_1 \sin \phi + \sqrt{l_2^2 - \left(l_1 \cos \phi\right)^2} - \sqrt{l_2^2 - l_1^2} \tag{1}$$

where $h$ is the displacement of the slider, $\phi$ is the rotation angle of the reduction gear, $l_1$ is the length of the crank, $l_2$ is the length of linkage.

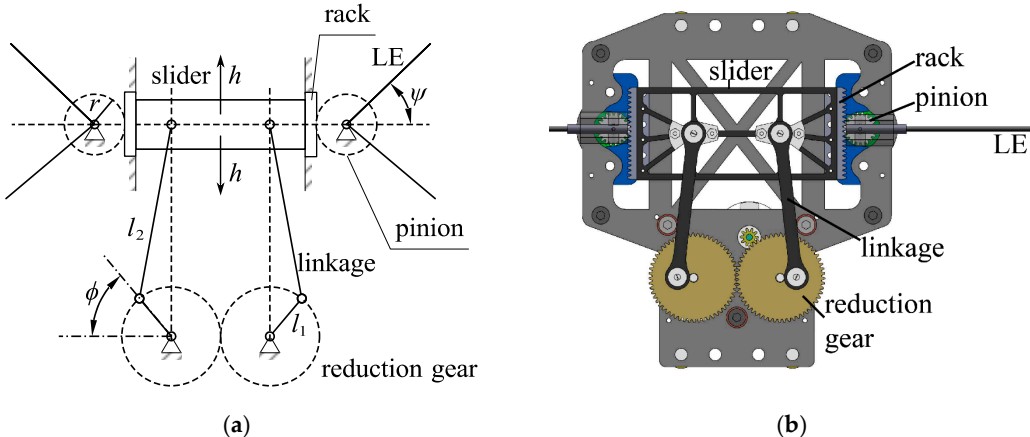

|  |  |
|---|---|
| (**a**) | (**b**) |

**Figure 1.** (**a**) Schematic of the flapping wing mechanism; (**b**) Assembled CAD model of the flapping wing mechanism.

Thus, the flapping angle $\psi$ of the leading edges (LEs) is given by:

$$\psi = h/r \tag{2}$$

where $r$ is the radius of the pinion graduation circle.

The LEs are made of carbon-fiber-reinforced polymer (CFRP) rods of 1.5 mm and were rigidly installed to the pinion gears to output the flapping motion. Therefore, the output flapping angle tends to be a cosine curve, approximately, as referred to in Figure 2, which represents the designed flapping angle in one flapping cycle.

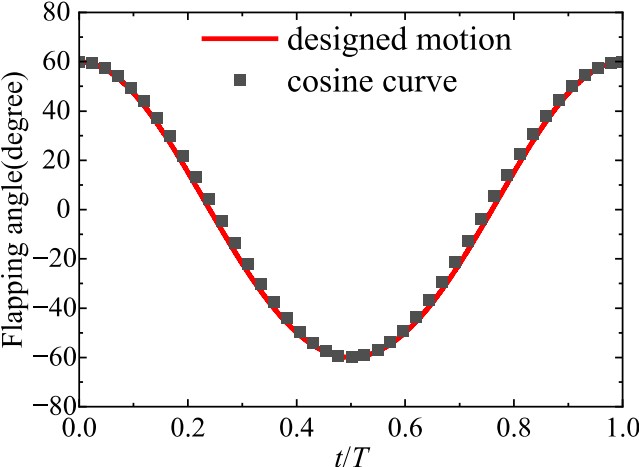

**Figure 2.** The designed output flapping angle of the mechanism compared with cosine curve. $T$ is the flapping period, and $t$ is the physics time.

The main structure of the mechanism is made of organic glass by laser cutting. The flapping frequency is controlled by adjusting the input current of the DC motor. An oscilloscope was used to measure the flapping frequency, which analyzed the output signal of a Hall sensor mounted on the mechanism. The Hall sensor could generate a rising-edge signal once the magnet embedded in the reduction gear passes through the Hall sensor. The maximum achievable flapping frequency of the flapping mechanism is almost 30 Hz. In this paper, the flapping angle is fixed at 120°. Notably, due to the flexibility of the leading edge rod, the actual flapping amplitude might increase with the increase of frequency.

### 2.2. Wing Design

In this study, the wings presented in Figure 3 were designed to perform a hovering flapping motion with passive twisting. To test the influence of wing membrane materials on their aerodynamic performance, three different membranes with similar thicknesses were introduced, and their properties are shown in Table 1. Meanwhile, the shape of these wings is identical. These wings have an area of 4500 mm$^2$ and aspect ratio (AR) of 2.4. As there is an offset between the wing root and the flapping axis, the tip radius $R$, from wing tip to the flapping axis, equals 120 mm, the radius of the second moment of wing area $R_2$ can be expressed as [22]:

$$R_2 = \sqrt{\frac{1}{S}\int_0^R cr^2 dr} \tag{3}$$

where $S$ is the wing area, $r$ is the distance from the flapping axis to any radial position along the wingspan, and $c$ is the local chord length at the given radial position $r$. For these three wings, the $R_2$ is the same value and equals 70.8 mm. Consequently, the non-dimensional radius of the second moment of wing area $\hat{r}_2 = R_2/R$ equals 0.59, which is similar to the insect wing parameter in nature [23]. The Reynolds number is given by:

$$Re = \frac{U_{ref}\bar{c}}{v} \tag{4}$$

where the reference velocity $U_{ref} = 4f\psi_m R_2$, $f$ is the flapping frequency, $\psi_m$ is the flapping amplitude and is set to 60°, and $\bar{c}$ is the mean chord length and is given by $S/R$, $v$ is the kinematic viscosity of air. As the flapping frequency in this experiment ranges from 10 Hz to 22 Hz, the Reynold number $Re$ varies from 7650 to 16,900.

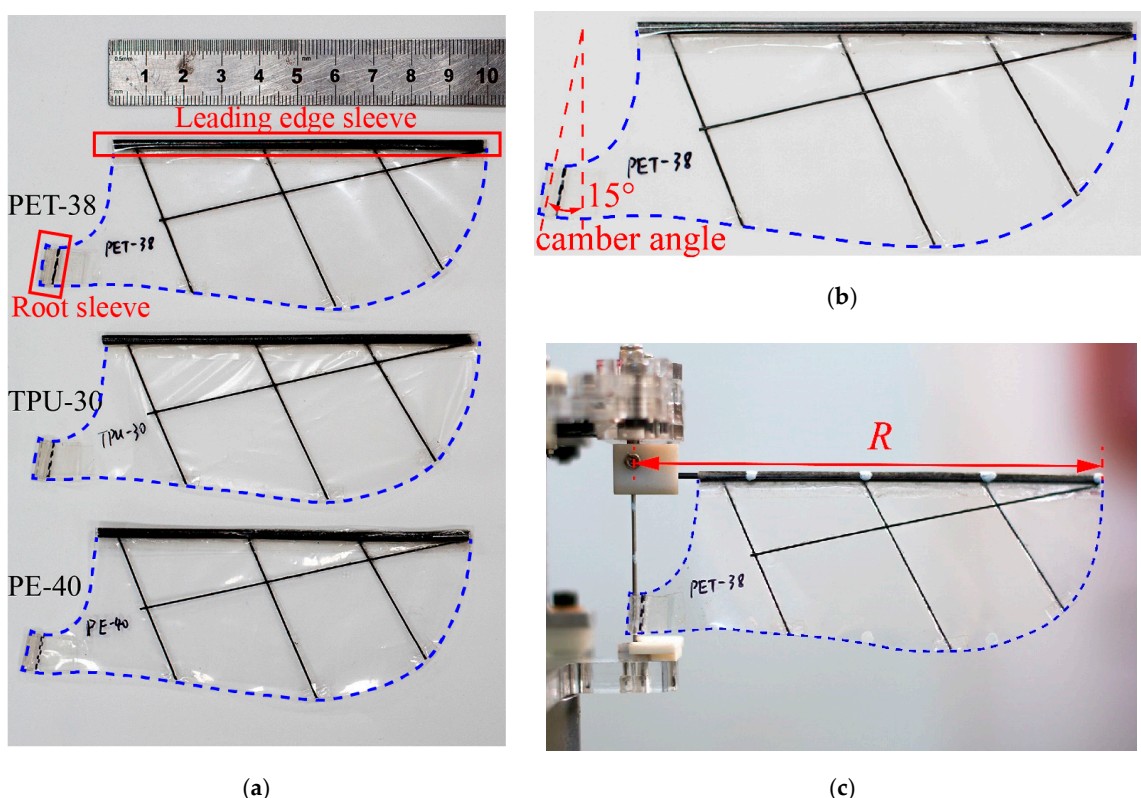

**Figure 3.** Wings designed for the test. (**a**) Photograph of three wings; (**b**) Description of the camber angle; (**c**) Wing assembled on the mechanism.

**Table 1.** Material properties of the tested Wings.

| Wing Material | Thickness (μm) | Elastic Modulus (GPa) | Density (g/cm³) |
|---|---|---|---|
| Polyethylene terephthalate (PET) | 38 | 3.30 | 1.38 |
| Polyethylene (PE) | 40 | 1.72 | 0.92 |
| Thermoplastic urethanes (TPU) | 30 | 1.08 | 1.10 |

Wing planform consists of membrane and CFRP stiffeners. The wing surface is stiffened by carbon bars of 0.5 mm in diameter to maintain the shape of the wing during its flapping period. Of note, the CFRP stiffeners are placed at the same location for all three wings. An expanded polystyrene glue is used to bond CFRP rods and wing membranes together.

The wing contains two sleeves, the leading edge sleeve and the wing root sleeve. The diameter of the leading edge sleeve is about 2 mm, so the wing can rotate freely around the leading edge CFRP rod. A steel bar with a diameter of 1 mm is inserted through the sleeve of the wing root to fix the wing root. A camber angle, defined as the angle between the wing root edge and a normal to the leading edge, is introduced to achieve the ideal wing twisting deformation. In this study, the camber is set to 15°, which has been proven to be better for the aerodynamic performance [24].

## 3. Data Acquisition and Deformation Measurement

### 3.1. Lift and Power Measurement

A series of tests were conducted for these three wings under various flapping frequencies. Figure 4 presents the measurement setup. The lift force was defined as the vertical force generated by the flapping wing. During the force measurement, the flapping wing mechanism was connected to a six-axis force/torque sensor (Mini40, ATI Industrial Automation Inc., Apex, NC, USA), which was attached to the support. The force resolution of this load cell is 1/50 N. An in-house LabView program was used to record the force and torque data through a data-acquisition board (National Instruments, Austin, TX, USA), with a sampling rate of 2000 Hz. The program was designed to record the force data within the next 1 s after each trigger, and at least 10 flapping cycles' data were recorded according to the flapping frequency set in this research. Subsequently, more than 20 force measurements have been conducted during the stable flapping process at each flapping frequency. Meanwhile, the fifth-order low-pass Butterworth filter was adopted to filter the force data. The cut-off frequency is three times the flapping frequency, which has been proven to be effective in filtering high-frequency vibrations of the mechanisms while retaining reliable force data [25].

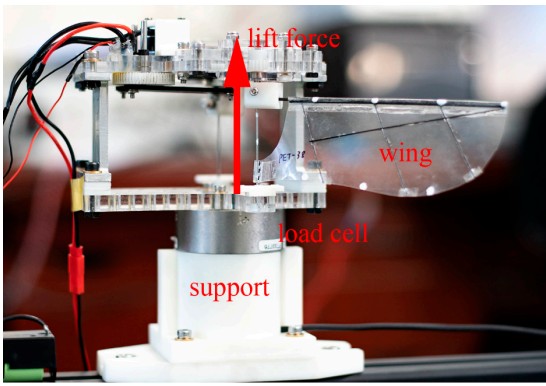

**Figure 4.** Flapping wing mechanism attached to the support.

The flapping mechanism was excited by an external DC power supply, and the output voltage of the power supply is set to a constant value of 8.4 V. Since the flapping frequency

of the wing was controlled by adjusting the input current, the power input $P_{in}$ of the mechanism can be expressed as:

$$P_{in} = UI \tag{5}$$

where $I$ is the input current.

### 3.2. Deformation Measurement

The rotation angles of three different wing sections (25%, 50%, and 75% of the wingspan) are measured to obtain the wing's deformation. As shown in Figure 5a, the white-colored markers with a diameter of about 2 mm are placed at the leading edge and corresponding trailing edge. The flapping process is filmed with two synchronized high-speed cameras (Phantom VEO-E 340L) with a resolution of 1280 × 960 pixels at 2000 fps. As presented in Figure 5b, the cameras are arranged approximately orthogonally to clearly capture the marked points. The position of the markers was tracked manually in each frame using the open-source digitizing tool, DLTdv8, developed by Hedrick [26], and based on the direct linear transformation method. Meanwhile, two additional markers were placed on the steel bar fixed at the wing root, which were tracked as a reference to calculate the rotation angle.

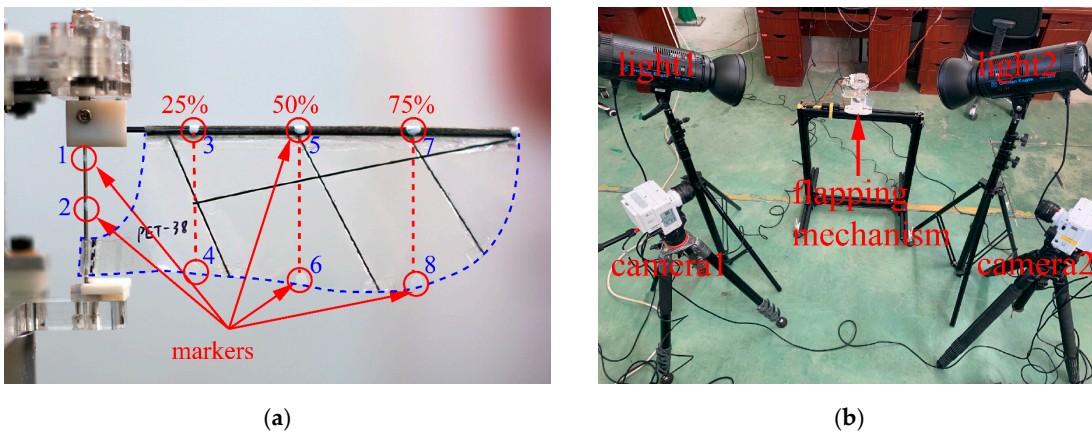

|   (**a**)   |   (**b**)   |

**Figure 5.** Deformation measurement setup. (**a**) Markers on the wing; (**b**) The experimental layout for the deformation measurement.

As seen in Figure 6, the chord line is the line connecting the leading and trailing edge at a specified wingspan section. The rotation angle $\theta$ is defined as the angle between the chord line and the vertical line, which is parallel to the flapping axis. The angle of attack $\alpha$ is defined as the angle between the incoming flow and the chord line. According to the geometric relationship, the angle of attack $\alpha$ can be expressed as:

$$\alpha = \pi/2 - \theta \tag{6}$$

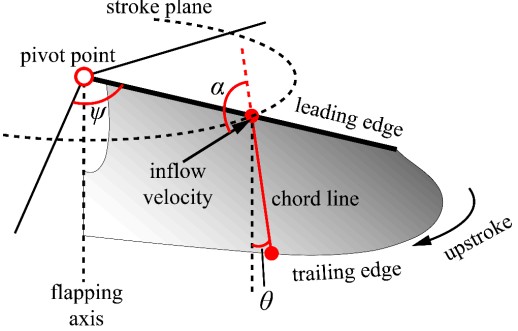

**Figure 6.** Definition of the flapping angle variables.

## 4. Results and Discussion

### 4.1. Lift and Power

Figure 7 presents the measured results for these three wings at various flapping frequencies. The force data was presented in the form of mean value with the corresponding standard deviations. It should be noted that due to the limitation of DC power supply equipment, the displayed current is always the same and at constant value during the test under a given flapping frequency, and does not fluctuate in any way. Thus, the standard deviation of the power input was not presented in Figure 7b.

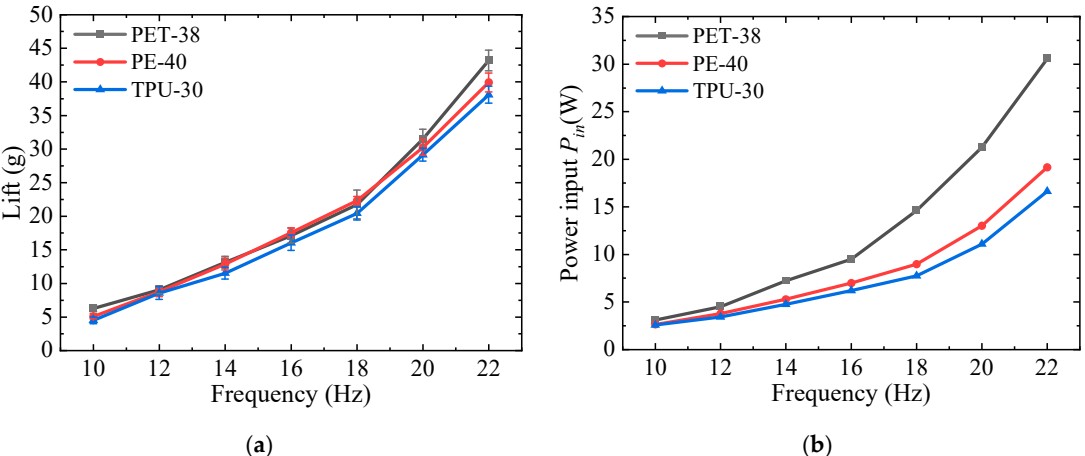

(**a**)                                            (**b**)

**Figure 7.** Measured force and power input for the three wings at various flapping frequencies. (**a**) Mean lift force; (**b**) Power input.

It can be found that the three lift curves present a similar trend. As the flapping frequency increases, the mean lift increases monotonically. Meanwhile, at a higher flapping frequency, the lift grows faster, implying a nonlinear relationship between the lift and flapping frequency. By referring to the research of Nan et al. [17] and Nguyen et al. [27], the mean lift might be proportional to the second power of the flapping frequency.

For all flapping frequencies, the lift of TPU-30 is the smallest among the three wings. When the flapping frequency is lower than 18 Hz, the lift of PET-38 and PE-40 are basically the same. While at a higher flapping frequency, PET-38 starts to present an advantage in lift generation. Moreover, the higher the frequency, the more obvious the advantages.

According to Nguyen et al. [27], who analyzed the power consumption of the flapping wing system, the power input can be mainly decomposed into aerodynamic power, mechanism power, inertia power, etc. In this study, the power of each component has not been separated. Although comparing the aerodynamic power for these wings is better for evaluating the aerodynamic performance, the power consumption of the flapping system still needs more attention. The power input, which means the power required to drive the wing's movement, can be regarded as the power consumption for the flapping system.

As shown in Figure 7b, similar trends in the three power input curves can also be observed, which increase with the increase of flapping frequency. At a higher flapping frequency, the power input grows rapidly, implying a greater power required at a higher frequency. More precisely, the relationship of power input against flapping frequency also seems nonlinear. This phenomenon can also be found from the research by Nan et al. [17], who conclude the power consumption scaled with approximately a third of the power of the frequency. Similar to the result of lift, the input power of PET-38 is the largest of all flapping frequencies, while that of TPU-30 is the smallest. The higher the frequency, the greater the difference in power consumption of the three wings. At 22 Hz, the power consumed by the PET-38 is almost twice as much as that consumed by the TPU-30.

Power loading, defined as the lift generated by per unit power (Lift/$P_{in}$), can be considered as the efficiency of power input conversion into lift generation, and is shown

in Figure 8. The calculation of the standard deviation for the power loading follows the error propagation criterion [28]. Of note, different from the result of the lift and power input, the power loading curves of the three wings present an inconsistent trend. With the increase of the flapping frequency, the power loading of PET-38 decreases monotonously, and the maximum value appears at 10 Hz. The power loading of PE-40 increases first to the peak value at 16 Hz, after which it drops gradually. The power loading of TPU-30 gradually increases and then decreases, and the maximum value is obtained at 20 Hz. In other words, each wing has an optimum flapping frequency to obtain maximum efficiency, which is consistent with the research by Godoy-Diana and Thiria [12]. It can be found that the power loading of TPU-38 is the largest among most flapping frequencies, while that of PET-38 is the smallest, although this wing has the best lift performance.

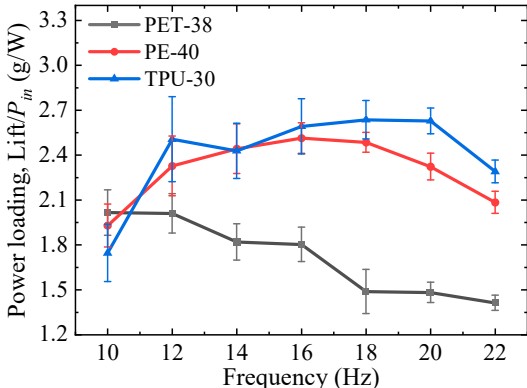

**Figure 8.** Power loading versus frequency for different wings.

By referring to the material prosperities of the three wings membrane, it can be found that wing flexibility has a significant influence on lift generation and power consumption performance. A lower elastic modulus membrane means larger flexibility of the wing. Under the same wing configuration, the PET-38 with a higher elastic modulus membrane presents better lift performance with more power consumption. This phenomenon suggests that flexibility may be detrimental to lift generation. However, a more flexible wing shows advantages in energy saving. Consequently, a more flexible wing achieves higher power loading, even the lift performance is not that outstanding. Meanwhile, as the wing becomes more flexible, the flapping frequency where the wing obtains the peak power loading increases. The result suggests that the flexible wing works more efficiently at a higher frequency.

### 4.2. Deformation Results

Using the high-speed cameras and DLT method, the rotation angles of three sections ($0.25R$, $0.5R$, $0.75R$) of the wings are measured and calculated to analyze the deformation characteristics. Of note, as the flapping process of wings is periodic and repetitive, it can be assumed that the rotation angle is repeatable in each cycle. Therefore, only the rotation angle in one flapping period is presented in the subsequent sections.

#### 4.2.1. Effect of the Flapping Frequency

Figure 9 shows the rotation angles of PET-38 in a flapping period at the aforementioned wing sections at the flapping frequency of 14 Hz. Apparently, the rotation angle turns to be larger along the wingspan, indicating that the twisting deformation on the outboard wing is more significant. The transition from downstroke to upstroke is accompanied by the dramatic change of the rotation angle, along with continuous fluctuation. The fluctuation of the rotation angle can be cleanly seen in Figure 9b, during the wing's translation phase. Different from the research by Nguyen et al. [20], who found that the rotation angle is almost constant during the translation phase, the reason might be the flapping angle (120°)

is smaller than that adopted by Nguyen et al. [20]. Hence, before the rotation angle's fluctuation turns to be attenuated, the wing rapidly flaps in a reverse direction.

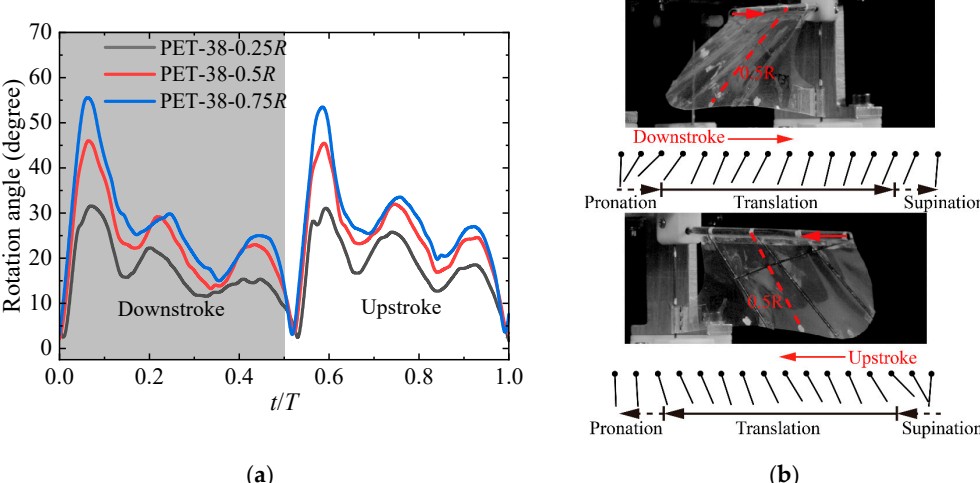

(**a**)                                                                                     (**b**)

**Figure 9.** Wing deformation of PET-38 at 14 Hz. (**a**) The rotation angles of PET-38 measured at three wing sections. The gray zone indicates that the wing is flapping downstroke; (**b**) Diagram of wing motion drawn based on the rotation angle at 0.5*R*; the black dot means the leading edge.

It can be found that the rotation angles at three wing sections show similar trends, suggesting similar deformation characteristics on the wing. Meanwhile, a slight phase shift between the three wing sections can also be found, which seems like a deformation transmission along the wingspan. This phenomenon is more obvious for more flexible wings at high frequencies. The phase shift between different wing sections was also found by Heathcote et al. [29], on a heaving flexible rectangular wing. It should be noted that there is a slight deviation in the rotation angles between the down and upstroke phases. The reason might be asymmetric rotation caused by the assembly of the wing root sleeve and the installation tolerance of the gearbox.

The effect of flapping frequency on the rotation angle is shown in Figure 10. The flapping frequency will significantly affect the reverse process and fluctuation of the rotation angle. As the flapping frequency increase, the proportion of transition from downstroke to upstroke turns to be larger. During the translation phase, fluctuation times decrease, and the timing of the second peak rotation angle is delayed. Additionally, the flapping frequency mainly affects the deformation on the outboard wing. The rotation angle on the outboard wing is larger at a higher frequency.

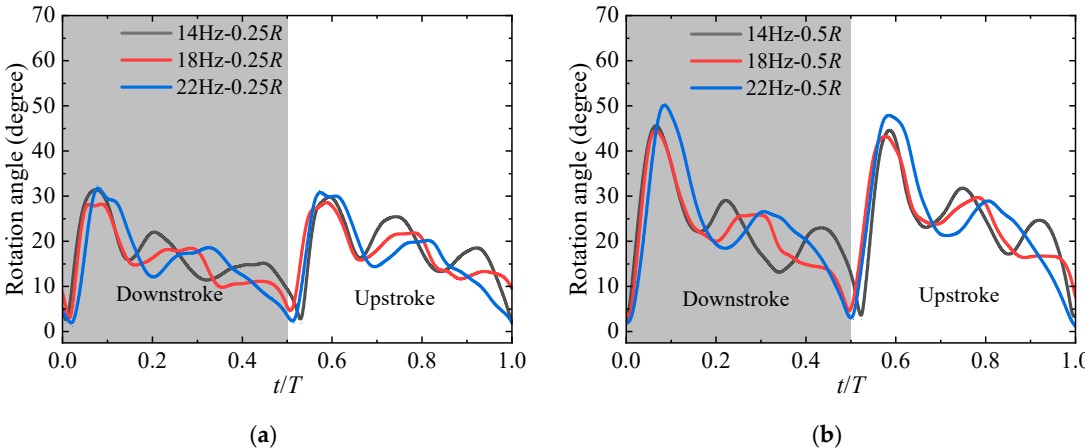

(**a**)                                                                                     (**b**)

**Figure 10.** *Cont.*

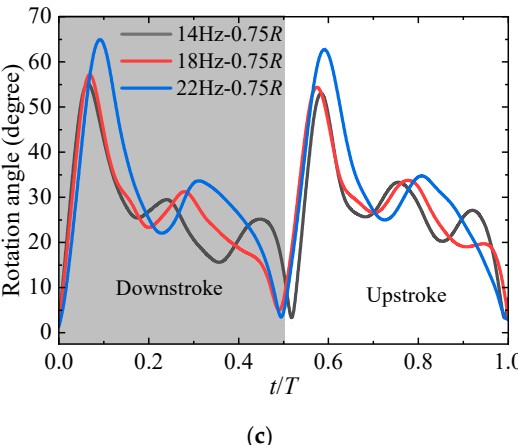

(**c**)

**Figure 10.** Wing deformation of PET-38 at 14 Hz, 18 Hz, and 22 Hz. (**a**) The rotation angles at 0.25*R*; (**b**) The rotation angles at 0.5*R*; (**c**) The rotation angles at 0.75*R*.

The effect of flapping frequency on the wing deformation can also be seen from the cycle-averaged rotation angle presented in Figure 11. The cycle-averaged rotation angle was calculated by averaging the rotation angle over a flapping period, and can be defined as:

$$\overline{\theta} = \frac{1}{T} \int_{t}^{t+T} \theta(t) dt \tag{7}$$

where *T* is the flapping period. With the increase of the flapping frequency, the deformation along the spanwise direction turns to be larger.

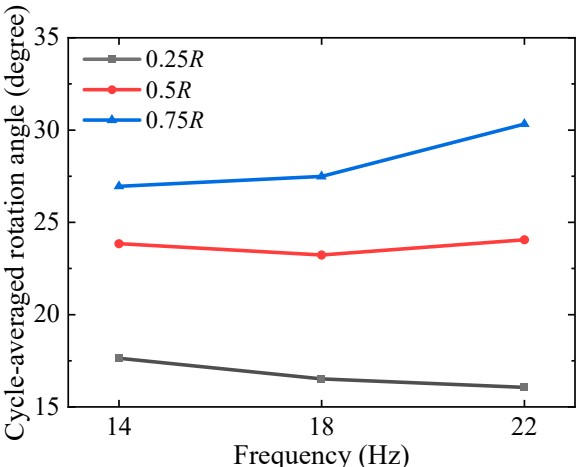

**Figure 11.** Cycle-averaged angle of PET-38 at different flapping frequencies.

4.2.2. Effect of Wing's Flexibility

In this section, considering the similar lift performance of PET-38 and PE-40, only the deformation of PET-38 and TPU-30 is presented.

The rotation angles of PET-38 and TPU-30 are presented in Figure 12. It can be found that at almost all flapping frequencies, more flexible TPU-30 shows a larger rotation angle than PET-38. Furthermore, the amplitude of angle fluctuation seems smaller for the TPU-30, suggesting that flexibility plays a dissipative role in the angle fluctuation. As referred to in Figure 8, TPU-30, with slighter angle fluctuation, obtained higher power loading, suggesting that eliminating the fluctuation is beneficial for improving efficiency, which is consistent with the research of Lee et al. [21].

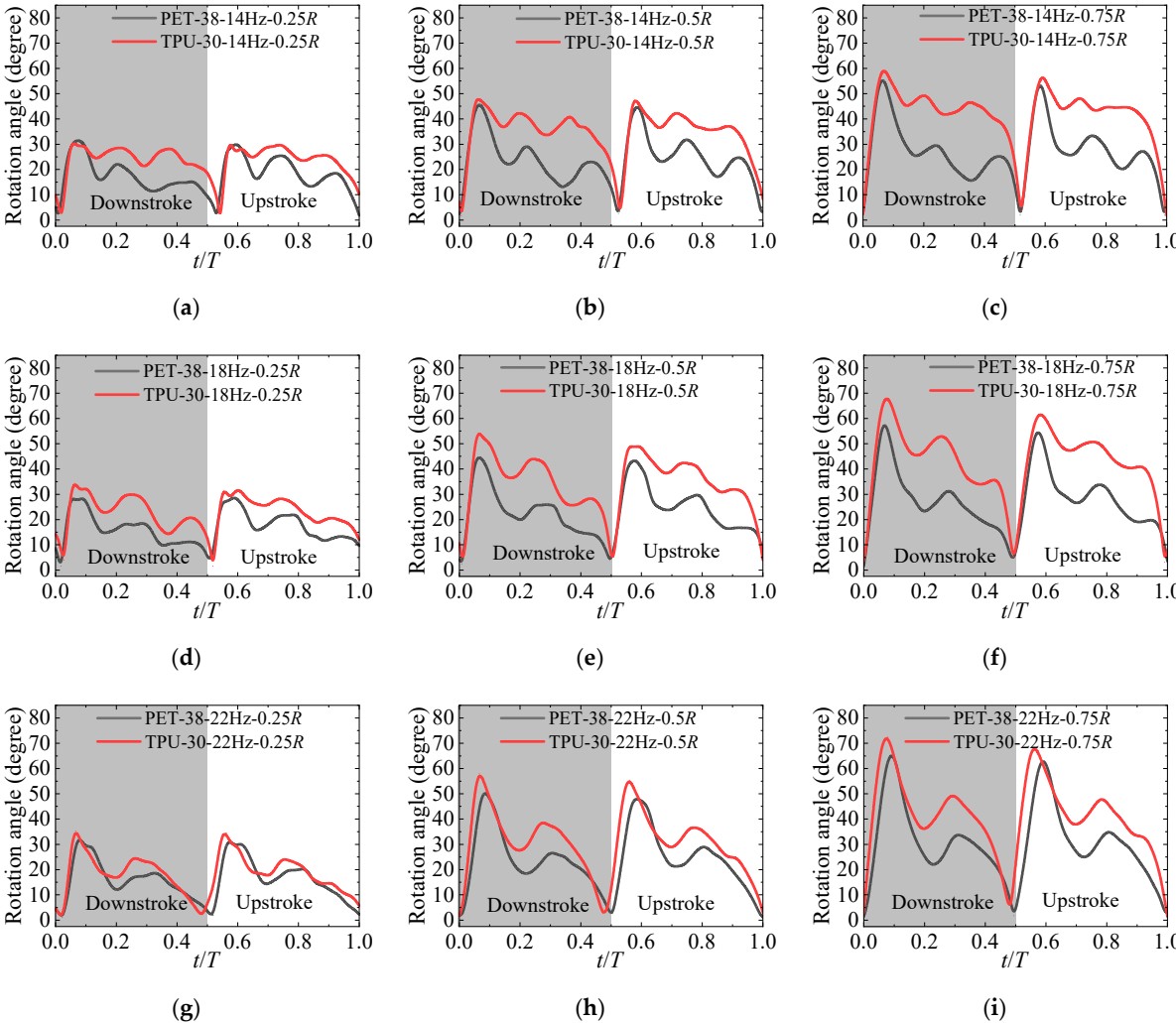

**Figure 12.** Deformation comparison of different wing sections at: (**a**) 0.25*R* of 14 Hz; (**b**) 0.5*R* of 14 Hz; (**c**) 0.75*R* of 14 Hz; (**d**) 0.25*R* of 18 Hz; (**e**) 0.5*R* of 18 Hz; (**f**) 0.75*R* of 18 Hz; (**g**) 0.25*R* of 22 Hz; (**h**) 0.5*R* of 22 Hz; (**i**) 0.75*R* of 22 Hz.

Apparently, even if the wing membrane materials are different, the fluctuation trends of PET-38 and TPU-30 are still similar. According to Lee et al. [21], the vein position was able to affect the angle fluctuation and the aerodynamic efficiency. As the identical vein configuration was adopted for the PET-38 and TPU-30, it can be concluded that the effect of membrane material on the angle fluctuation trend is less obvious than that of wing vein.

Differences in wing deformation with different flexibility can also be reflected by the cycle-averaged angle shown in Figure 13. Obviously, the cycle-averaged rotation angle of TPU-30 with lower elastic modulus membrane is significantly larger than that of PET-38. The increase in rotation angle along the spanwise direction also explains the twisting deformation of TPU-30, which is more prominent at a higher frequency.

It can be seen from Figure 13b that PET-38 has a larger cycle-averaged angle of attack, which is close to 65 degrees, whereas the angle of attack of TPU-30 ranges from 20 degrees to 60 degrees. As presented in the study by Dickinson et al. [1], the optimal translation component of lift can be obtained at an angle of attack of 45 degrees, based on the quasi-steady aerodynamic model, as seen in Figure 14. Under this circumstance, the lift of TPU-30 should be larger than that of PET-38 at a certain flapping frequency, since the angle of attack of TPU-30 is closer to 45 degrees, as seen in Figure 13. However, by looking at Figure 7a, PET-38 with a larger angle of attack obtained a higher lift. This contradiction

implies that there is another noticeable lift component that contributes to the resultant force apart from the translation lift.

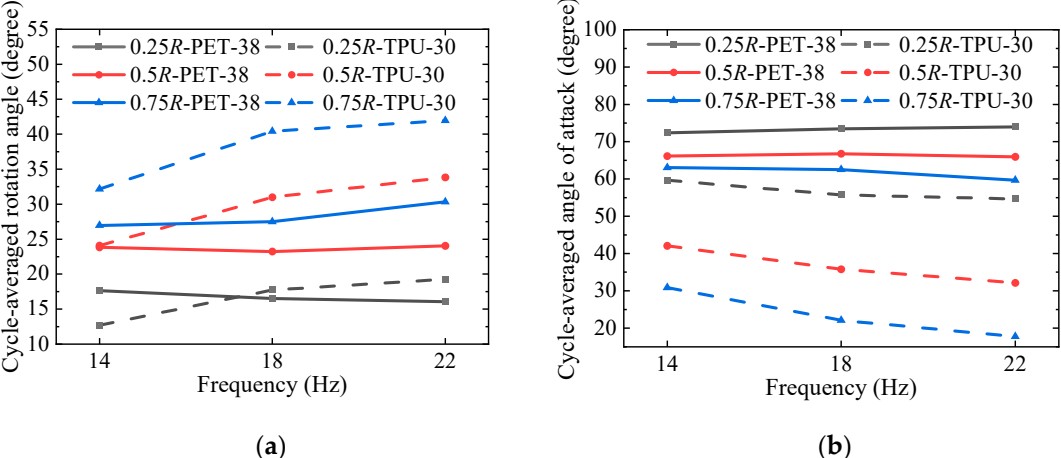

**Figure 13.** Cycle-averaged angle comparison of PET-38 and TPU-30. (**a**) Cycle-averaged rotation angle; (**b**) Cycle-averaged angle of attack.

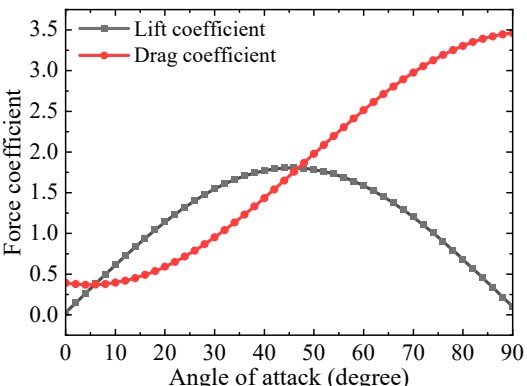

**Figure 14.** Translational lift and drag coefficients calculated based on the research by Dickinson et al. [1].

According to the quasi-steady aerodynamic model developed by Lee et al. [22] and Addo-Akoto et al. [30], instantaneous forces can be represented as the sum of translational force, rotational force, and added mass force. Rotational force and added mass force are closely related to the flapping kinematics, especially angular velocity and angular acceleration of the angle of attack at a given spanwise blade element. Since the flapping kinematic is identical for the same flapping frequency, the difference in lift force for the two wings might be attributed to the discrepancy in the angle of attack caused by different flexibility.

In the research of Phan et al. [31], the lift is primarily generated by the outboard wing during its hovering flight. Thus, angular velocity and angular acceleration of the angle of attack at 0.75*R* for 22 Hz are analyzed and presented in Figure 15. According to the quasi-steady aerodynamic model aforementioned, the rotational force is the function of the angular velocity of the angle of attack, and the added mass force is formulated by the angular acceleration.

Although the trend of angular velocity and angular acceleration curves of PET-38 and TPU-30 are similar, there are still differences in values, especially during the stroke reversal. At this phase, PET-38 obtained a larger positive angular velocity and angular acceleration than TPU-30, thus PET-38 might have a larger rotating force and additional mass force, which will eventually lead to a higher total lift. Additionally, as seen in Figure 14, a larger angle of attack of PET-38 will lead to a higher drag force, which is defined in the stroke

plane and opposite to the flapping direction. Higher drag force will contribute to more energy consumption, corresponding to a higher power input for PET-38, as referred to in b.

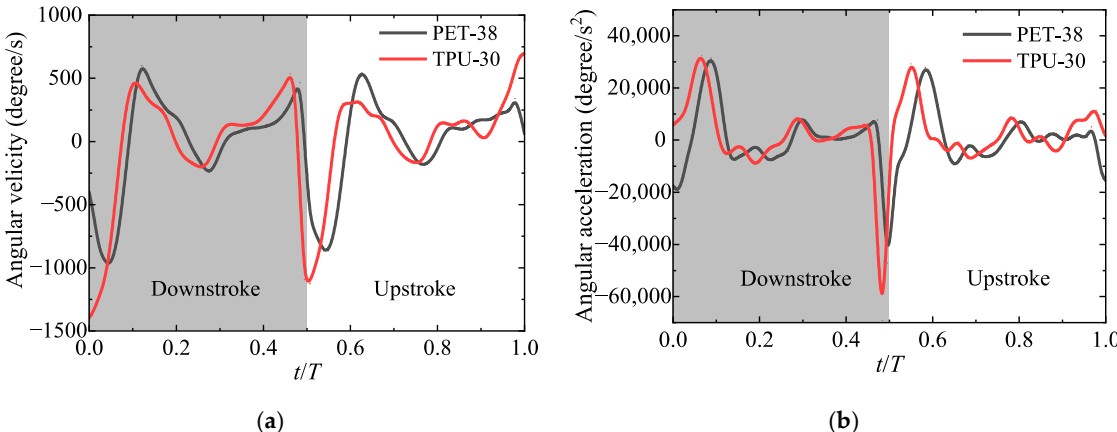

**Figure 15.** Angular velocity and angular acceleration of the angle of attack at 0.75*R* for 22 Hz. (**a**) Angular velocity; (**b**) Angular acceleration.

## 5. Conclusions

In this study, the aerodynamic and deformable characteristics of flapping wings with different wing membrane materials have been experimentally investigated. Three materials (PET, PE, TPU) with similar thicknesses were selected as the wing membrane to fabricate the tested wings. Lift force and power input were measured during the hovering flight. Rotation angles at different wing sections were analyzed to evaluate the deformation process. The following conclusions are drawn from the study:

(1) Flexibility presents a significant influence on lift generation and power consumption. Wings with a higher elastic modulus membrane, which means a more rigid wing, could generate higher lift but at the cost of more power.

(2) Wings with a lower elastic modulus membrane present greater twisting deformation along the spanwise direction, especially at a high flapping frequency. Meanwhile, the twisting deformation is more obvious with the increase of flapping frequency.

(3) A more flexible wing shows an advantage in translational force for a smaller angle of attack, close to the optimal value. However, the potentially superior rotational force related to higher rotation angular velocity and angular acceleration may be responsible for the higher lift of a more rigid wing, yet a larger angle of attack also forms a larger drag force and greater power consumption.

(4) Rotation angle fluctuation occurs after the stroke reversal. Flexibility shows advantages in attenuating the rotation angle fluctuation, which in turn enhances the aerodynamic efficiency.

**Author Contributions:** Conceptualization, B.S.; Formal analysis, X.L.; Funding acquisition, X.Y.; Writing—original draft, X.L.; Writing—review & editing, W.Y. All authors have read and agreed to the published version of the manuscript.

**Funding:** This research was funded by the National Natural Science Foundation of China, grant number11872314, 11902103; the Key R&D Program in Shaanxi Province of China, grant number 2020GY-154; and the foundation of National Key Laboratory of Science and Technology on Aerodynamic Design and Research.

**Institutional Review Board Statement:** Not applicable.

**Informed Consent Statement:** Not applicable.

**Data Availability Statement:** Not applicable.

**Conflicts of Interest:** The authors declare no conflict of interest.

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
