# Peer review of "Effect of Wing Membrane Material on the Aerodynamic Performance of Flexible Flapping Wing"

_applsci, doi:10.3390/app12094501_

Round 1
Reviewer 1 Report
The paper presents an interesting study of flexible wing performance. The topic is well presented and the conclusions are significant. Small corrections are needed: - line 53: please define FMWAV here in text as well (not just in abstract) - line 257: please use either quite small or smaller - line 255: Maybe "Obtained results differ from the results of the research-- " would be better In conclusion please indicate the direction of further research (for example, wind tunnel experiment)Author Response
Thanks for your helpful suggestions and advice. We have answered the comments one by one. Please see the attachment.

Reviewer 2 Report
The article is almost complete - indicated minor corrections and additions - comments in the attachment.

Author Response
Thanks for your helpful suggestions and advice. We have answered the comments one by one. Please see the attachment.

Reviewer 3 Report
I believe that the article is worthy of publication, ready for publication and I recommend to the Editorial Board to publish it.

Author Response

(The authors gave the same response as above.)
